# Validation of a Sapphire Gas-Pressure Cell for Real-Time In Situ Neutron Diffraction Studies of Hydrogenation Reactions

Raphael Finger [1] , Thomas C. Hansen [2] and Holger Kohlmann [1],*

1 Institute of Inorganic Chemistry, Leipzig University, Johannisallee 29, 04103 Leipzig, Germany; raphael.finger@uni-leipzig.de

2 Institut Laue-Langevin, 71 Avenue de Martyrs, 38000 Grenoble, France; hansen@ill.fr

* Correspondence: holger.kohlmann@uni-leipzig.de; Tel.: +49-341-9736201

**Abstract:** A gas-pressure cell, based on a leuco-sapphire single-crystal, serving as a pressure vessel and sample holder, is presented for real time in situ studies of solid-gas hydrogenation reactions. A stainless steel corpus, coated with neutron absorbing varnish, allows alignment for the single-crystal sample holder for minimizing contributions to the diffraction pattern. Openings in the corpus enable neutron scattering as well as contactless temperature surveillance and laser heating. The gas-pressure cell is validated via the deuteration of palladium powder, giving reliable neutron diffraction data at the high-intensity diffractometer D20 at the Institut Laue-Langevin (ILL), Grenoble, France. It was tested up to 15.0 MPa of hydrogen pressure at room temperature, 718 K at ambient pressure and 584 K at 9.5 MPa of hydrogen pressure.

**Keywords:** in situ neutron diffraction; gas-pressure cell; hydrogenation; reaction pathways





## 1. Introduction and Nomenclature

Hydrogen is an important resource for many industrial processes, e.g., the synthesis of ammonia or methanol. Nowadays, hydrogen also gets a lot of attention for its applications as an energy carrier. With the continuous transition towards 'green energies', several fields like photovoltaic or hydroelectricity advanced strongly in recent years [1,2]. Mostly, these energies are considered as 'green', meaning they are carbon neutral in contrast to fossil energy carriers, like mineral oil and coal. As they are not constantly available due to the sun or tides, suitable energy storage systems have to be found. The development of storage materials with high capacity and long-term stability is one of the key areas towards energy transition and green energy, and puts a high demand on new materials.

As the International Energy Agency reports, the energy demand will increase heavily in the next two decades [3]. While fossil fuels produce climate gases like $CO_2$ upon burning and are a limited resource, hydrogen burns to just water and its availability is virtually unlimited. This has been acknowledged in politics and led the German government to release a proper hydrogen strategy in which the strong support for the production and usage of hydrogen and its infrastructure is described [4]. For these reasons, hydrogen is viewed as a top choice to tackle the rising energy demand, due to its availability and carbon neutrality. To enable an effective energy distribution, the energy needs to be stored outside of sun or tide cycles and be released on-demand. Several types of hydrogen storage are being used such as physical [5] (compressed gas and liquid $H_2$), physicochemical (adsorption of $H_2$ molecules in porous materials) and chemical [6] (incorporation of H atoms or $H^-$ ions in chemical compounds). Focusing on inorganic compounds, intermetallic compounds like Fe–Ti [7] or $LaNi_5$ [8] are able to absorb and desorb hydrogen at moderate conditions, however, suffer from low weight efficiency [8,9]. In the last two to three decades, so called light-weight materials were introduced, such as alanates, boranates, amides, complex hydrides and metal organic frameworks (MOFs) [9,10]. Alanates have a high hydrogen

capacity, for example $Mg(AlH_4)_2$, with 9.3 wt.% but suffer from high temperatures necessary to release the full hydrogen content. Similarly, $LiBH_4$ has a high hydrogen capacity of about 13.4 wt.% but a high dehydrogenation temperature of 653 K [11,12].

To understand functional materials under operating conditions, in situ and *operando* techniques are important [13–17]. In our contribution, we use the term in situ to describe the constant collection of data while changing several physical parameters, e.g., temperature and/or pressure, according to [18]. To study light elements like hydrogen, neutron scattering as a characterization technique is suitable due to its sensitivity as compared to e.g., X-rays. Therefore, hydrogen can be located in a crystal structure containing heavy elements, as being an established method for more than 70 years [19,20].

To give a further understanding of hydrogenation reactions in general, in situ diffraction of hydrogen uptake and release is essential. In this respect, improved sample environment such as gas-pressure cells are in demand. This contribution is about further developments of a sapphire single-crystal gas-pressure cell, called type I cell, for real-time neutron diffraction [18,21]. The main difference of the type II cell described here as compared to the type I cell is a direct connection between the top and bottom side of the sample holder, allowing a more homogenous distribution of clamping stress and an easier assembly and handling. It therefore offers a high potential for future improvements, as higher gas pressure for example. This work will give basic details about development, operating conditions, reliability, and assembly of type II sapphire gas-pressure cells for in situ neutron powder diffraction investigations of solid–gas reactions. Since the main field of application so far was hydrogenation reactions, the validation of this new type of gas-pressure cell was done by in situ neutron diffraction studies of the palladium–deuterium system. Sapphire crystals used for technical purposes are colorless and chemically very pure, and therefore sometimes called leuco-sapphire instead. For reasons of simplicity and to follow the customs of the scientific community, however, we call these colorless corundum crystals sapphire.

The herein presented gas-pressure cell is a versatile tool for many kinds of solid-gas reactions. Virtually any not-corrosive gas may be used instead of hydrogen and deuterium and it may easily be adapted to gas flow instead of static gas pressure. This makes it an interesting opportunity for the in situ observations of chemical reactions of solids in many areas of materials-related research (material sciences, physics, chemistry) and also for industrial processes.

## 2. Assembly and Use of the Single-Crystal Gas-Pressure Cell (Type II)

As centerpiece of the type II gas-pressure cell, the sapphire single-crystal sample holder from the type I cell are used [18]. It is a Kyropoulos grown cylindrical crystal (along crystallographic c-axis) with a height of 100 mm, having flanges on top and bottom. The sample is placed in a borehole with a diameter of 6 mm, being drilled into the crystal. In the type II sapphire single-crystal gas-pressure cell the base mount from the type I cell is replaced by a stainless steel corpus (violet, Figure 1, and Appendix A) made out of one piece of austenitic chromium-nickel stainless steel (EN 1.4301/AISI 304). The flange joint (Figure 1, green), similar to the one of the type I cell, is screwed to the corpus, in order to seal it for gas-pressure. In contrast to the type I cell, the flanges are pressed on from one side each only, which helps distributing clamping stress more uniformly. Further, the assembly is easier. For better understanding, Figure 2 shows a vertical cut view of the type II cell as well as a horizontal cut along the diffraction plane, showing several openings which are needed to perform in situ neutron scattering experiments. On top of the corpus, six ISO metric screw threads M6 with suitable flat washer and hex nuts serve as connection between corpus and flange joint, allowing gas-tight sealing. As seals on the top flange and buffer material on the bottom flange, polymeric flat seals NBR 65 and FKM 70 [13] have been successfully used. To fix the type II cell at the neutron diffractometer, an M8 borehole is added on the bottom side of the corpus.

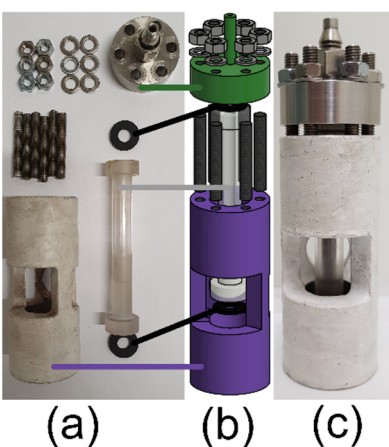

**Figure 1.** Type II gas-pressure cell: (**a**) disassembled, (**b**) exploded view, (**c**) assembled; green: flange joint, violet: corpus, grey: sapphire sample holder, black: seal.

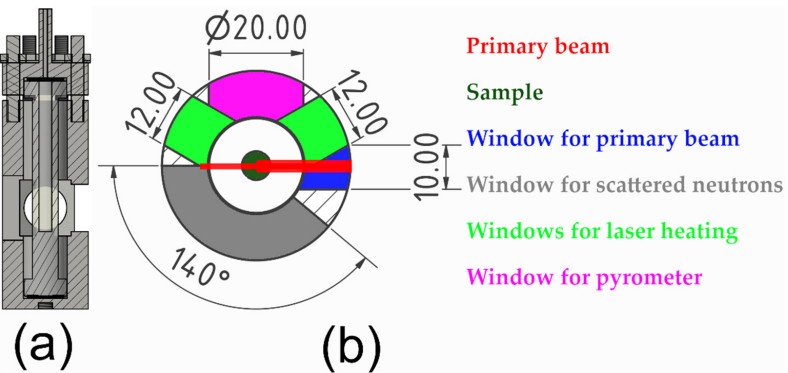

**Figure 2.** Type II sapphire gas-pressure cell: (**a**) vertical cut view, (**b**) cut along diffraction plane, length specification in mm.

Several windows are cut into the corpus (Figure 2), allowing for a free optical pathway with 140° visible for the neutron detector. The window for the primary beam is 10 mm in width and 22 mm in height. Two windows of 12 mm width and 26 mm height, being ranged in a 120° angle allow for laser heating via two water-cooled laser diodes (980 nm, 100 W, LNT, Großumstadt, Germany) and another one for temperature surveillance via a calibrated pyrometer [18]. The window for scattered neutrons is 26 mm high. An increased height of the window did not show improved data quality. The stability of the corpus was tested for 8 weeks using a sample holder out of aluminium being fixed by screwing the nuts with a torque of 5 Nm. No mechanical fractures or irregularities of the corpus were observed.

To prevent neutron activation and parasitic reflections of the stainless-steel corpus, it is coated by a neutron absorbing and temperature stable varnish. 1 mL of a heat resistant and transparent varnish spray (Dupli-Color Supertherm 500), usually used for engine parts, and 5 g of well-ground gadolinium oxide, $Gd_2O_3$ powder (99.9%, Rudolf Gerresheim, Lübeck), are mixed. The resulting dispersion is painted on the corpus and dried for 24 h in air at room temperature. This procedure is repeated three to four times until the corpus is covered homogenously with a neutron absorbing varnish. Cadmium metal sheets are wrapped around the flange joint during in situ neutron diffraction experiments, in order to prevent neutron activation.

To assemble the type II leuco-sapphire gas-pressure cell, three main parts are needed: sample holder (grey, Figure 1 right), corpus (violet, Figure 1 right) and flange joint (green, Figure 1 right). Flat polymer rings (NBR 65 or FKM 70) are placed in the corpus as washer material and on top of the sapphire single-crystal as gas-pressure seal (black, Figure 1

right). It is important to use the same material as buffer and seal to ensure homogeneous distribution of clamping stress along the sample holder to prevent damage. Then, the screw threads are connected to the corpus and the flange joint is carefully put on top. Finally, the nuts are fixed crosswise in incremental steps up to 1.2 Nm to ensure equal pressure distribution to prevent jamming of the sample holder. Detailed technical drawings may be found in the Appendix A. For the flange joint, the drawings are already shown in [18], supporting information. So far, we restricted test conditions to what was needed for our in situ studies (Table 1). Therefore, higher temperatures and pressures may be reached using this gas-pressure cell.

**Table 1.** Tested gas-pressure ($p$) and temperature ($T$) conditions for the type II gas-pressure cell.

| Wall Thickness/mm | Gas Pressure/MPa | $T/K$ |
|---|---|---|
| 3 | 0.1 air | 571 |
| 3 | 2.5 $H_2$ | 524 |
| 3 | 5.0 $H_2$ | 435 |
| 3 | 15.0 $H_2$ | 298 |
| 2 | 0.1 air | 524 |
| 2 | 15.0 $H_2$ | 298 |
| 1 | 0.1 air | 718 |
| 1 | 9.5 $H_2$ | 298 |
| 1 | 9.5 $H_2$ | 584 |

The openings in the corpus limit the cell to a certain orientation in contrast to the type I cell. In order to test suitable angular ranges for turning the cell around its tube axis ($\varphi$ or $\omega$), an $\omega$-scan was performed. Silicon powder, Johnson and Matthey, 325 mesh, 99.999%, was put into an aluminum crucible instead of a sapphire crucible. During an $\omega$-scan of 20° in 1° steps, 30 s per measurement, no significant changes in intensity were observed (Figure 3). However, two parasitic reflections could be seen at about 120° and 128° in 2θ for $\omega \geq 16°$. This experiment shows that the cell may be turned around $\omega$ by about ±8° without severing the diffraction pattern, which is by far enough for fine tuning $\omega$ positions as for minimizing sample holder contributions to the pattern. Type II cells are thus suitable for the same kind of in situ neutron diffraction setup as described for type I [13].

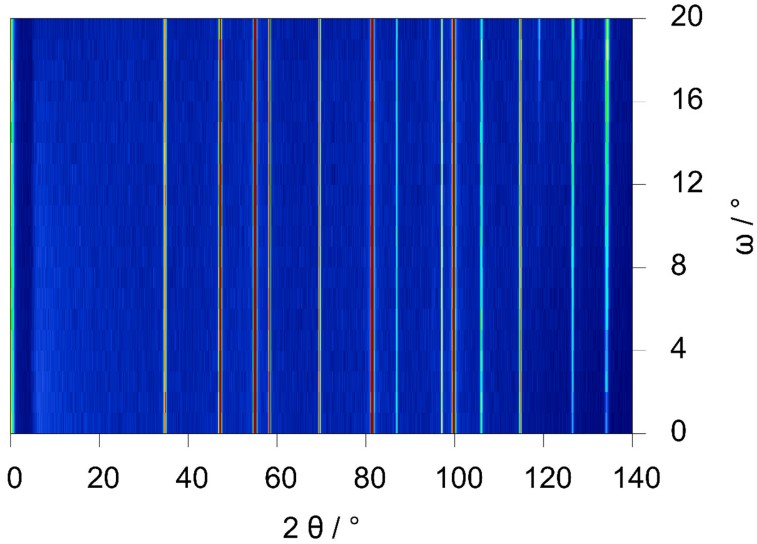

**Figure 3.** 2D-plot of an $\omega$-scan of silicon powder in an aluminium crucible performed on D20 at ILL in 1° steps of $\omega$ (corresponding to one slice) and 30 s data accumulation per slice in false colors, NUMOR 138781, doi:10.5291/ILL-DATA.5-24-621 with NUMOR being the internal raw data labeling of the Institute Laue-Langevin, Grenoble, France.

### 3. Validation of the Type II Sapphire Gas-Pressure Cell

In accordance with the validation of the type I cell [22,23], the type II cell, presented in this work, was validated via the deuteration of palladium powder. The Pd-D system was ideally suited due to the high symmetry of all phases and good scattering lengths of both constituent atoms. Deuterium was chosen instead of hydrogen due to its much lower incoherent scattering to prevent unwanted background contribution during the in situ experiment. Rietveld refinements were performed using pseudo-Voigt function for profile fitting in FullProf [24]. For the visualization of crystal structures, VESTA [25] was used.

*3.1. Experimental Details*

Palladium powder (99,95%, mesh size <150 micron, GoodFellow) was pretreated at $10^{-6}$ MPa and 773 K for 24h and was handled under argon afterwards before filling into the type II sapphire single-crystal gas-pressure cell. The in situ setups for type I and type II cells were identical [18]. The gas-pressure cell was evacuated (0.01 MPa) and the temperature raised to 435(1) K before slowly adding injections of deuterium up to 5 MPa (Figures 4 and 5a). During the deuteration, the unit cell of palladium was expanded by the incorporation of deuterium, occupying octahedral voids, see Figure 4b. As expected, the occupation of tetrahedral voids as seen above the critical point of the phase diagram [26] was not observed here since our in situ study was made way below the critical point.

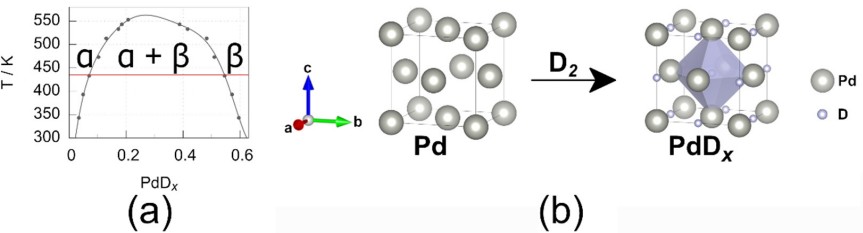

**Figure 4.** (**a**) Palladium–deuterium phase diagram based on absorption isotherms, data from [27] (grey points) fitted with polynomial function (grey line) and the temperature used for validation of the gas-pressure cell (red line), (**b**) visualisation of the deuteration of palladium with deuterium occupying the octahedral voids of the cubic close packing of palladium.

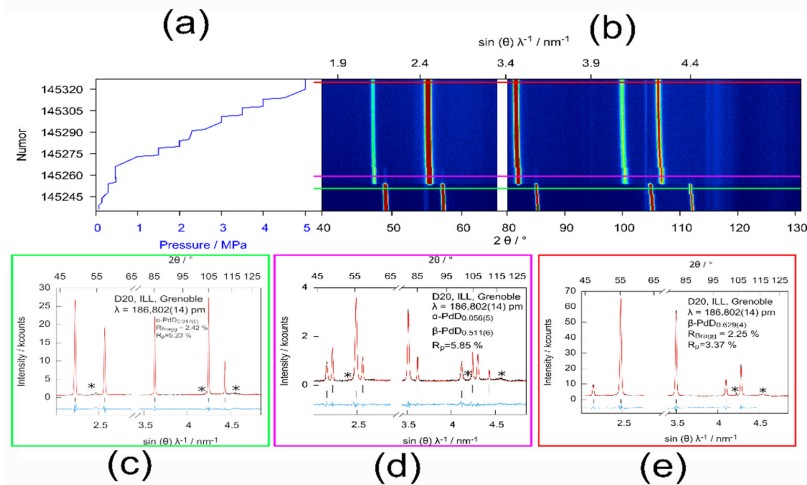

**Figure 5.** In situ neutron powder diffraction of the reaction of palladium powder with deuterium: (**a**) pressure profile, (**b**) diffraction data as false colour plot, (**c**) Rietveld refinement for $\alpha$-PdD$_{0.047(6)}$ (NUMORs 125247-145252; green line (**b**) and green box (**c**)), (**d**) Rietveld refinement for $\alpha$-PdD$_{0.056(5)}$ and $\beta$-PdD$_{0.512(6)}$ (NUMOR 145254; violet line (**b**) and violet box (**d**)), (**e**) Rietveld refinement for $\beta$-PdD$_{0.629(4)}$ (NUMORs 145317-145327; red line (**b**) and red box (**e**), parasitic reflection from sample environment marked with *, 2 min per NUMOR, doi:doi.org/10.5291/ILL-DATA.5-22-767 with NUMOR being the internal raw data labeling of the Institute Laue-Langevin, Grenoble, France.

### 3.2. Results and Discussion

In the course of the in situ experiment, $\alpha$- $PdD_x$ started to form immediately after the first deuterium injection. By further increasing the deuterium pressure, a second, hydrogen-richer phase, $\beta$-$PdD_x$, was formed. Finally, after crossing a two-phase region with both $\alpha$- and $\beta$-phase, only the latter was observed (Figure 5). Full crystal structure refinements were performed for all data sets. This allows the comparison of the refined deuterium content with that extracted from the phase diagram. Experimentally determined deuterium contents were in good accordance with literature data (Figure 6, Tables 2 and 3), proving the suitability of this type II cell for detailed crystal structure investigations using in situ neutron diffraction. Compared to our previous validation [18] of the type I cell, the data were in good accordance ($\alpha$-$PdD_{0.056(5)}$ at 433(1) K and 0.5 MPa as compared to $\alpha$-$PdD_{0.038(4)}$ at 446(4) K and 0.3 MPa).

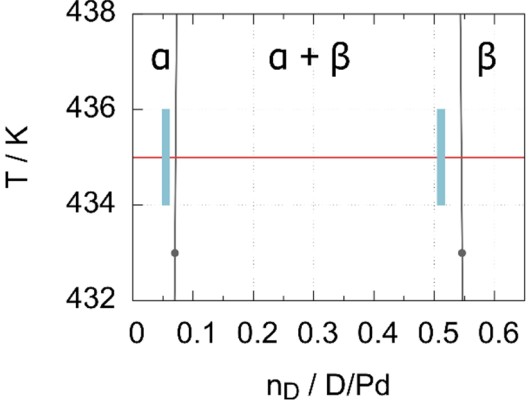

**Figure 6.** Part of the palladium–deuterium phase diagram (blown-up section of Figure 4a at 0.6 MPa deuterium gas pressure, data from [23], experimentally determined deuterium content at 0.5 MPa for the $\alpha$-$PdD_{0.056(5)}$ and $\beta$-$PdD_{0.512(6)}$ as well as temperature as red line and with $\pm \sigma$ as blue boxes).

**Table 2.** Crystal structures of palladium deuterides, $PdD_x$, at 435 K, between 0.3 MPa and 5.0 MPa $D_2$ pressure (space group type $Fm\overline{3}m$, Pd in $4a$ 0 0 0, D in $4c$ $\frac{1}{2}$ $\frac{1}{2}$ $\frac{1}{2}$).

| Phase | $p(D_2)$/MPa | Lattice Parameter/Å | Deuterium Content $x$ | $B_{iso}$(Pd)/Å$^2$ | $B_{iso}$(D)/Å$^2$ |
|---|---|---|---|---|---|
| $\alpha$-$PdD_x$ | 0.3 | 3.91016(19) | 0.047(6) | 1.17(5) | 4(1) |
| $\alpha$-$PdD_x$ | 0.5 | 3.91285(13) | 0.056(5) | 0.97(8) | 4 * |
| $\beta$-$PdD_x$ | 0.5 | 4.03114(15) | 0.512(6) | 1.24(6) | 3.9(1) |
| $\beta$-$PdD_x$ | 5.0 | 4.05067(22) | 0.629(4) | 1.44(4) | 4.54(6) |

* Fixed to the value of $\alpha$-$PdD_{0.047(6)}$ at 0.3 MPa.

**Table 3.** Comparison of the reflection/background ratio $R_I$ of all silicon reflections measured from neutron powder diffraction data using the type I and type II gas-pressure cell.

| hkl | Position $2\theta$/° | $R_I$ Type I | $R_I$ Type II-A | $R_I$ Type II-B |
|---|---|---|---|---|
| 111 | 34.80 | 27.3(2) | 24.5(2) | 23.8(3) |
| 220 | 58.35 | 35.0(3) | 29.7(2) | 32.1(3) |
| 311 | 69.70 | 32.3(3) | 26.8(3) | 29.7(4) |
| 400 | 87.10 | 10.5(1) | 8.7(1) | 10.3(2) |
| 331 | 97.25 | 18.9(2) | 16.0(2) | 18.6(2) |
| 422 | 114.85 | 30.5(2) | 25.2(0) | 29.8(3) |
| 511/333 | 126.65 | 15.6(3) | 15.5(2) | 15.2(2) |

### 3.3. Data Quality of the Type I and Type II Cell

To evaluate the data quality, neutron diffraction measurements of powder samples in type I and type II cells at ambient conditions were compared (Figure 7). For ensuring

optimal comparability, several requirements had to be met as for sample, sample holder and instrumental setup:

- The same sample (417.4(1) mg of silicon powder, (Johnson and Matthey, 325 mesh, Alfa Products, Karlsruhe, Germany), filling height 20 mm) was used for all measurements.
- The same sample holder in the same orientation has been used for all measurements, i.e., contributions from the sample holder were identical.
- All measurements were performed using the same diffractometer setup and were performed within 24 h.

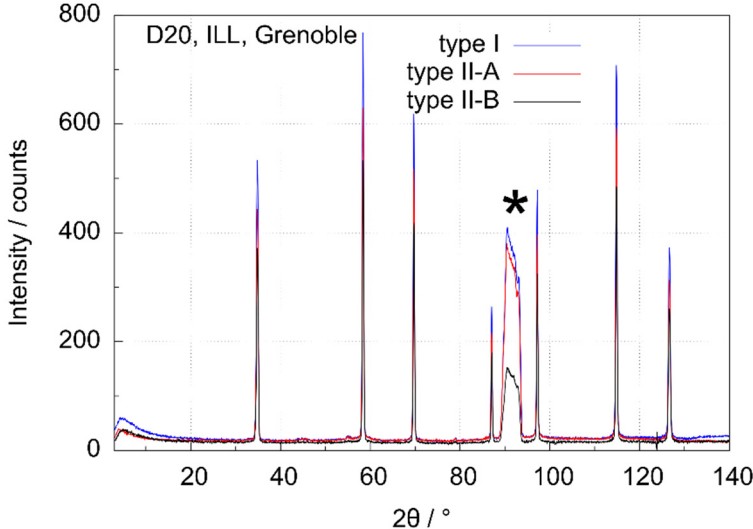

**Figure 7.** Neutron powder diffraction data of silicon powder in the type I, II-A and II-B cells at ambient conditions, sample holder contribution marked with *, D20, ILL Grenoble, NUMOR 178776, 178781, 178810, doi:10.5291/ILL-DATA.5-24-639 with NUMOR being the internal raw data labeling of the Institute Laue-Langevin, Grenoble, France.

For the type I cell, the sapphire single-crystal with the sample was mounted on the diffractometer using the base mount presented in [18]. The type II-A cell represents the corpus which was used for the validation reaction described above. Additionally, a type II-B corpus, having a smaller height for primary beam and scattered neutrons (20 mm instead of 26 mm for the type-II-A cell) while merging the horizontal cut for scattered neutrons and the primary beam, i.e., having a larger window to enable high angle neutron scattering up to the detector maximum of 153.6° 2θ, was tested.

$$\mathbf{R_I} = \frac{\mathbf{I_{max}} - \mathbf{I_{bgr}}}{\mathbf{I_{bgr}}}. \tag{1}$$

Formula to determine the reflection/background ratio for the Si reflections for the type I and type II gas-pressure cell, with $\mathbf{R_I}$ as reflection/background ratio, $\mathbf{I_{max}}$ as maximal reflection intensity, $\mathbf{I_{bgr}}$ as background

Reflection/background ratios for silicon reflections were within the same range (Table 1), but showed a slight decrease in the sequence type I–type II-B–type II-A cell. Thermal displacement parameters from Rietveld refinements, $B_{iso}$, are 0.55(2), 0.52(2) and 0.57(2) Å$^2$ at room temperature for type I, II-A and II-B, respectively, were in good agreement with literature data (0.54(1) Å$^2$) [28]. While the lower reflection/background ratios for measurements in type II cells (Table 3) were a disadvantage as compared to type I, reduced background was an advantage (Table 4). Especially at low diffraction angles this might facilitate the observation of weak superstructure reflections.

**Table 4.** Comparison of the average background between the different types of gas-pressure cells and corpora.

| Type | Average Low Angle Background/Counts ($5° \leq 2\theta \leq 25°$) | Average High Angle Background/Counts ($130° \leq 2\theta \leq 140°$) |
|---|---|---|
| I | 29.8 | 24.5 |
| II-A | 21.4 | 17.4 |
| II-B | 21.0 | 16.3 |

## 4. Further Developments

As the sample holder material has a high mechanical strength, temperatures and pressures may be pushed to much higher values then presented in this work. During our experiments, we focused on the available conditions, as limited by our heating device and the used gas-handling system. One of the advantages of this type of gas-pressure cell is its versatility and the potential for adaption for other uses, e.g., low temperature, other gases or gas flow instead of static gas pressure, all of which were not tested yet. This would open up new possibilities for studying important materials-related reactions, e.g., gas storage in porous materials, ore smelting processes, corrosion of metals or solid-state syntheses. For all such further applications the sealing will probably have to be adapted and optimized for each case. This is a crucial issue for the usability of the device and for the safety of its use, since gas leaks may be a serious hazard. Therefore, using such gas-pressure cells outside the conditions reported here should be done with utmost care and only after extensive safety tests adapted to the chosen conditions.

## 5. Conclusions

A sapphire single-crystal gas-pressure cell (type II) for in situ neutron diffraction has been developed and validated. It contains a cylindrical single-crystal sample holder having two flanges and a borehole to place the sample, which is held by a corpus machined from one piece of steel. The corpus is painted with neutron absorbing varnish and displays several windows for primary beam, scattered neutrons, contactless laser heating, pyrometric temperature measurement and optical surveillance. The gas-pressure cell has been validated via the deuteration reaction of palladium powder. The reflection/background ratio of type II-A and II-B cells are about 14% lower for type II-A and 6% for type II-B in comparison of the type I cell, while the overall background at low diffraction angles is reduced by 28% and 30%, respectively. At high angles, the reduction is 29% and 33% for the type II-A and II-B, respectively. The main advantages of the type II cell are easy handling, reduction of mechanical stress on the sapphire sample holder and a lower background at high and low angles, at cost of slightly lower reflection/background ratios as compared to the type I cell.

**Author Contributions:** Conceptualization, R.F. and H.K.; methodology, R.F., T.C.H. and H.K.; software, R.F. and T.C.H.; validation, R.F. and T.C.H.; formal analysis, R.F.; investigation, R.F., T.C.H. and H.K.; resources, H.K.; data curation, R.F.; writing—original draft preparation, R.F.; writing—review and editing, R.F., T.C.H. and H.K.; visualization, R.F.; supervision, H.K.; project administration, H.K.; funding acquisition, H.K. All authors have read and agreed to the published version of the manuscript.

**Funding:** This work was supported by the BMBF—German Federal Ministry of Education and Research [grant number 05K16OL1], and the Deutsche Forschungsgemeinschaft [grant number Ko1803/4-1].

**Data Availability Statement:** The DOI to the diffraction data and experiments are given in the caption of the corresponding Figure (Figures 3, 5 and 7). Additionally, the CIF Files (crystallographic information file) for the crystal structures of palladium deuterides at non ambient conditions, mentioned in Table 2, are available through the CCDC (The Cambridge Crystallographic Data Centre) via the IDs CSD 2081926, 2081927, 2081928 and 2081929, respectively.

**Acknowledgments:** We acknowledge the Institute Laue Langevin for provision of beam-time at the high-intensity powder diffractometer D20. Further, we acknowledge Henry Auer who took part in developing the concept used for the corpus of the gas-pressure cell.

**Conflicts of Interest:** The authors declare no conflict of interest.

**Appendix A**

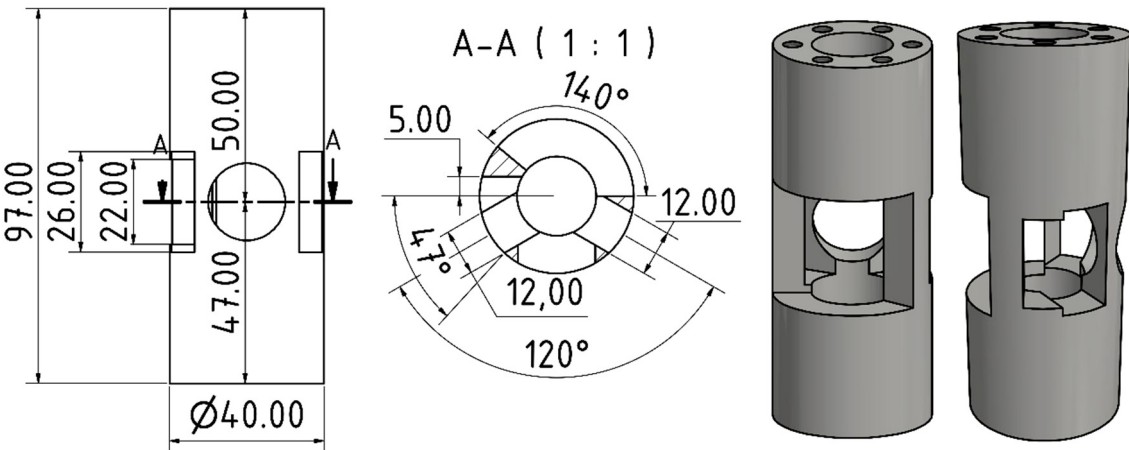

**Figure A1.** Technical drawing of the corpus (type II-A) for the type II leuco-sapphire single-crystal gas-pressure cell (type II) machined from austenitic chromium-nickel stainless steel (EN 1.4301/AISI 304): back view of the corpus (left), cut along the diffraction plane (middle left), view approximately perpendicular to primary beam showing window for scattered neutrons (middle right) and view along primary beam (left), length specifications in mm.

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
