# Peer review of "Validation of a Sapphire Gas-Pressure Cell for Real-Time In Situ Neutron Diffraction Studies of Hydrogenation Reactions"

_qubs, doi:10.3390/qubs5030022_

Round 1
Reviewer 1 Report
This work describes the design and validation of a sapphire gas-pressure cell for real time in-situ neutron diffraction measurements. The authors do a nice job describing the design of the sample holder and the appropriate validation steps. It is recommended that this article be accepted pending major revisions, as it is unclear how this design is fundamentally different than previous sample holder designs. Once this is clarified in an acceptable manner, this article will be of interest to researchers in the neutron diffraction field.
Line 11: Change “allows to align…” to “allows alignment for…”.
Line 34: “Several types of hydrogen storage are being used, such as…” This section would be improved by adding additional references that describe the types of hydrogen storage in use, as well as why hydrogen is a top choice to tackle the rising energy demands.
Line 56: “The main difference of the type II cell described here as compared to the type I cell is a direct connection between the top and bottom side of the sample holder allowing a more homogenous distribution of clamping stress and an easier assembly and handling.” It is unclear how this is a significant difference from the type I design. At the minimum, a model or photo comparing the two should be included.
However, this leads to my biggest reservation about the work. Reference [14] describes the type I cell. It is accepted for publication, but this reviewer was unable to access it to determine how the type II cell is a drastic improvement from the type I. If neither the paper on the type I or II cells has been published yet, but both were submitted around the same time frame, what was the motivation for two short, separate articles as opposed to documenting the entire work (initial design and iterative improvements) in a single manuscript? The authors need to expand on the differences between the holder designs and why the work on the type I and II holders justify two separate publications.
Line 107: “The gadolinium oxide varnish shows a thermal stability up to 1123 K for 24h in air without deterioration of integrity, as tested in a chamber furnace.” Showing the data that proves this, or including a relevant reference, would strengthen this claim as opposed to asking the reader to take it on faith.
Line 165: “The occupation of tetrahedral voids as reported before [22] has not been observed here.” Why is that? Can the authors offer an explanation?
Line 177: “This allows to compare refined deuterium content with the phase diagram.” For correct grammar, reword to something like “This allows comparison between refined deuterium content and the phase diagram.”
Author Response
see file attached

Reviewer 2 Report
The authors present an improved design of a gas-pressure cell based on a single crystal, sapphire sample chamber for neutron powder diffraction at moderate pressures (<95 bar) and temperatures (~300C) and suitable for use with hydrogen gas. The type II cell presented in the manuscript implements a number of improvements over the authors' previous type I design (e.g., simplified design and assembly, wide angle for scattered beam, and lower background).
The manuscript is written clearly and succinctly. Citations are adequate. The example chosen to determine cell performance (PdHx) is appropriate. The data presented supports the conclusions. The manuscript will be a useful reference for users of the cell at the ILL or at other neutron facilities.
I have a few questions and comments for the authors to consider:
- The intended use for the cell is in situ chemical reactions involving hydrogen at elevated temperature and pressure. Sapphire is mechanically tough and chemically resistant. The authors mention testing the cell with H2 at 9.5 MPa and 598K. How long was the cell kept at these conditions? More generally, are there expected materials compatibility issues, especially for prolonged use at higher T and P?
- Could this cell design be used below room temperature (with a suitable flat gasket replacement if necessary)? Some materials used for hydrogen storage (physisorption) do rely on storage below room temperature.
- Interest in hydrogen storage has declined over the past decade, but use of the cell would be interesting beyond this narrow application. A broader array of inorganic and organic reactions involving not just hydrogen, but also many other gases of interest to a wide variety of scientists (chemists, material scientists, physicists,...) could be studied with this cell that has low background and is more chemically inert than other materials traditionally used in gas-pressure cells for neutron scattering. As they authors point out, the ability to perform time-dependent, in situ experiment is attractive and would be useful to chemists. Consider generalizing your introduction to include a broader array of potential customers and applications than just hydrogen storage.
Author Response
see file attached

Round 2
Reviewer 1 Report
The authors have sufficiently addressed previous comments and put together a nice manuscript that is now ready for publication.